# Palliative care service utilization and associated factors among cancer patients at oncology units of public hospitals in Addis Ababa, Ethiopia

**Nigus Afessa**[1], **Dagmawit Birhanu**[1], **Belete Negese**[1], **Mitiku Tefera**[2]*

**1** Department of Nursing, School of Nursing and Midwifery, Asrat Woldeyes Health Science Campus, Debre Berhan University, Debre Berhan, Ethiopia, **2** Department of Midwifery, School of Nursing and Midwifery, Asrat Woldeyes Health Science Campus, Debre Berhan University, Debre Berhan, Ethiopia

* mitikutefera2632@gmail.com

## Abstract

### Background

Palliative care helps patients and their families deal with the hardships that come with a life-threatening illness. However, patients were not fully utilizing the palliative care services provided by healthcare facilities for a number of reasons. In Ethiopia, there hasn't been any research done on the variables that influence the utilization of palliative care services.

### Objective

To assess palliative care service utilization & associated factors affecting cancer patients at public hospitals oncology units in Addis Ababa, Ethiopia.

### Methods

An institution-based cross-sectional study design was carried out. A structured and pre-tested questionnaire was administered to 404 participants at Tikur Anbesa Specialized Hospital and Saint Paul's Hospital Millennium Medical College from July 4 to August 2, 2022. A systematic random sampling technique was used to select the study participants. The data was collected by ODK-Collect version 3.5 software and exported to excel and then to SPSS version 25 for recoding, cleaning, and analysis. Logistic regression model was employed. P-values <0.05 were regarded as statistically significant.

### Result

About 404 participants' responded questionnaire giving a 97.6% response rate. The extent of Palliative care service utilization was 35.4% [95% CI: 31.4, 40.3%]. College or university education were 2.3 times more likely and living in a distance of <23 km from PC service centers were 1.8 times more likely to use palliative care services. Factors hindering palliative care service utilization were inability to read & write, treatment side effects, long distance to a health institution, and low satisfaction with the health care service.

data. But if you request at any time we will send any other data.

**Funding:** The authors received no specific funding for this work.

**Competing interests:** The authors have declared that no competing interests exist.

## Conclusion and recommendation

The extent of palliative care service utilization which was low. Factors to palliative care service utilization were clients' education level, treatment side effects, distance to a health institution, and patients' satisfaction. Interventions to enhance health education and counseling of cancer patients, early detection and management of treatment side effects and accessibility of palliative care services for cancer patients should be emphasized and implemented by all concerned stakeholders.

## Introduction

Palliative care (PC) refers to the active comprehensive care of a patient's body, mind, and spirit, and so is a powerful complement to the psychosocial care of the entire patient [1,2]. Throughout a serious illness, palliative care emphasizes symptom management and quality of life [3]. A good understanding of the fundamentals of successful communication, symptom management, and end-of-life care is critical regardless of whether the care is inpatient or outpatient [4]. PC includes symptom control and end-of-life management, as well as communication with families and the establishment of care goals that ensure dignity in death and decision-making power [5].

Over 29 million people died worldwide as a result of conditions that needed palliative care. Of these, 9.6 million deaths are new cancer-diagnosed cases. A total of 20.4 million people were predicted to require palliative care at the end of their lives. Adults make up 94% of those who require palliative care, with 69% being over 60 years old and 25% being between 15 and 59 years old. But 78% of those in need of palliative care live in low- and middle-income countries [6,7].

Cancer is a leading cause of morbidity and mortality worldwide, especially in developed nations. In the United Kingdom, one in every two people will be diagnosed with cancer at some point in their lives, with one in every four dying from it.

The global burden of cancer is expected to keep increasing, especially in developing nations. To prevent long referral processes and delays in the delivery of care, effective cancer treatment necessitates the availability of surgery, radiation, and therapy in the same location. Chemotherapy for cancer is not currently included on the Ethiopian Essential Medicines List. In the majority of public hospitals, even basic painkillers are difficult to obtain [8].

Palliative care services are unavailable in 42% of the world's countries. The critical shortage of palliative care services in low-resource settings results in significant personal, family, and societal expenses [9]. Most people view palliative care as a way to stop receiving life-saving treatments or to let them pass away. Moving consultation earlier in the hospitalization of "dying" patients is a bigger concern than encouraging more people to use palliative services earlier in the course of their disease [10].

Palliative care was provided to 50% of patients getting palliative radiation therapy (RT) for metastatic cancer, but it was underutilized in all patients receiving RT, particularly those with lung cancer and those treated in an outpatient environment [11].

Patients with cancer use palliative care at varying rates due to different factors [12]. According to a study done in Switzerland, patients' and families' misconceptions regarding palliative care serve as cognitive barriers to their use [13]. A research conducted in the United States found that among cancer patients treated as inpatients, PC services were used by 8.5% of patients throughout their stay [14].

**Predisposing factors**

- Demographic characteristics (Age, Sex, Educational status, Marital status , Religion, residence)
- Health beliefs factors (Knowledge, Attitude)

**Health Need factors**

- Cancer site &Severity of pain
- Treatment side effects



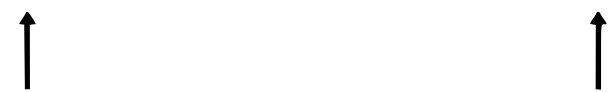

**Enabling factors**

- Family Income
- Occupational status
- Distance to health facility
- Satisfaction (health service, Information, Charity support, Family support )

**Health system factors**

- Health service fee
- Availability of medications and procedures
- Bureaucracy services
- Time and attention from health care provider

**Fig 1. Schematic presentation of conceptual framework of palliative care service utilization among cancer patients in Addiss Abeba, Ethiopia.**

There are different reasons that contribute to low palliative care service utilization among cancer patients in underdeveloped nations are multi-faceted, complicated, and little understood [15]. A study conducted in Ethiopia found that many cancer patients did not take advantage of palliative care services, and that patients with greater monthly incomes and families with more than two members were more likely to use these services [16]. Study's in Africa, especially in Ethiopia variables of hindering factors like treatment side effects and distance to cancer treatment centers during palliative care services did not incorporated. Therefore, the objective of this study was to assess palliative care service utilization status and factors affecting it among patients diagnosed with cancer at public hospitals oncology unit in Addis Ababa, Ethiopia (see Fig 1).

## Methods

**Study design, setting and period.** An institutional-based cross-sectional study was conducted. The study was conducted at Tikur Anbesa Specialized Hospital (TASH) and Saint Paul's Hospital Millennium Medical College (SPHMMC) which are located at the capital city of Ethiopia, Addis Ababa from July 4 to August 2, 2022.

## Population

**Source population.**   All adult patients diagnosed with cancer at Tikur Anbesa Specialized Hospital and Saint Paul's Hospital Millennium Medical College, Addis Ababa, Ethiopia, 2022.

**Study population.**   Selected adult patients diagnosed with cancer at Tikur Anbesa Specialized Hospital and Saint Paul's Hospital Millennium Medical College oncology unit, Addis Ababa, Ethiopia, 2022

## Eligibility criteria

**Inclusion criteria.**   All adult cancer patients (aged $\geq$ 18 years) who had been diagnosed with any type of cancer and on treatment follow-up before the data collection period were included in the study.

**Exclusion criteria.**   Those critically ill patients diagnosed with cancer who were non cooperative during the data collection period were excluded.

**Sample size determination and Sampling procedure.**   The sample size for this study was generated using a formula for a single population proportion based on the assumptions listed below. A 95% confidence level, the margin of error (0.05), a previous related study of proportion 0.572 [17] and by taking 10% non-response rate, then the total sample size required to this study was 414 cancer patients.

Both TASH and SPHMMC oncology center public hospitals were selected and done proportional allocation to each hospital based on their monthly number of patients on treatment follow-up. A systematic random sampling technique was used to select the study participants, and the registration log book of adult patients diagnosed with cancer was used as a sampling frame. The first participant was selected by lottery method, while the remaining individuals were selected by every two k intervals across both hospitals *(see Fig 2)*.

**Data collection methods.**   Two nurses and one supervisor were trained for the data collection. A standardized interviewer-administered questionnaire adapted from different literature [17–19] and modified in the current context was used to collect data. ODK-Collect software version 3.5 was used to collect data. The questionnaire was written in English following a thorough assessment of previously verified published research. Then, the questionnaire was translated to the Amharic language and the data collector used the Amharic version of the questionnaire to collect data from the study participants. Again, it was translated back into English for analysis.

## Study variables

**Dependent variable.**

➢ Palliative Care Service Utilization

## Independent variables

**Predisposing factors.**   Socio- demographic characteristics: age, sex, educational status, marital status, Religion and residence Health beliefs factors: knowledge and attitude

**Health Need factors**–perceived severity of illness, treatment side effects

**Enabling factors**–family income, occupational status, satisfaction, distance to health facility & transportation fee

**Health system factors**–health service fee, availability of medications and procedures, bureaucracy services, time and attention from health care provider

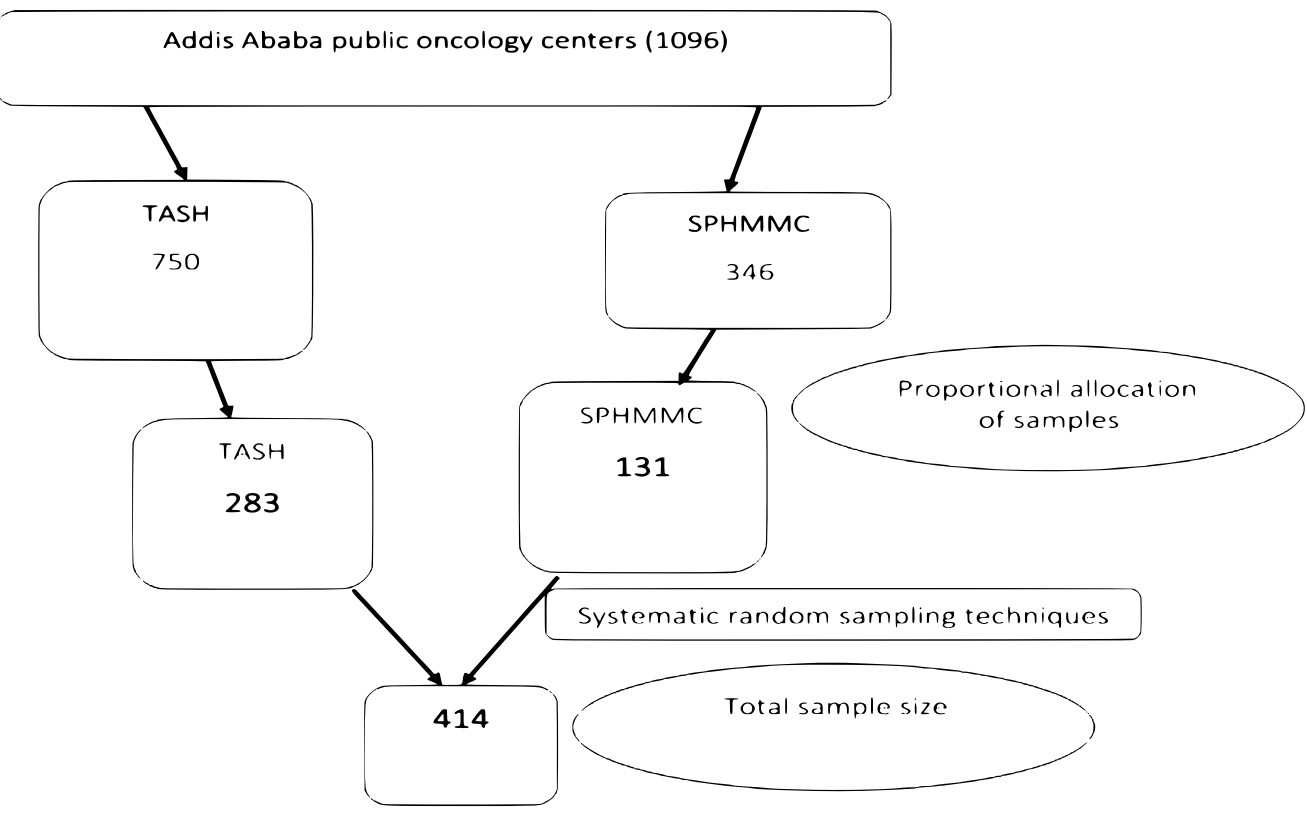

**Fig 2. Demographic presentation of sampling procedure on palliative care service utilization among cancer patients in Addis Abeba, Ethiopia.**

**Data quality management.** Data collectors and supervisors were trained in data gathering processes using ODK-Collect software to ensure that the data was of high quality. The questionnaire was properly designed and written in English first, then translated into Amharic by language experts, and finally back to English to ensure consistency. Before the questionnaire was used to collect data, it was pretested on 5% (21) of cancer patients at Dessie referral hospital and any necessary adjustments or modifications were made like language translation errors and coherence. On a daily basis, the main investigators have been evaluated and checked the obtained data for completeness and consistency. Finally, data was gathered using an Amharic version.

**Data processing and analysis.** Before data collection, each questionnaire was double-checked for accuracy and kobo Toolbox templet for ODK application data collection was coded and then collected data was exported to excel and from excel to SPSS version 25 for recoding, cleaning, and analysis. Bivariate logistic regression was utilized to look at parameters related to palliative care usage. To choose potential variables for multivariable logistic regression analysis, first a bivariate logistic regression analysis was employed. Bi-variable logistic regression variables with p-values <0.25 were incorporated into a multivariable logistic regression model to discover their independent correlation with the dependent variable. P-values <0.05 were regarded as statistically significant. The multivariable analysis results were presented as an adjusted odds ratio with a 95% confidence interval. Descriptive statistics were used to present socio-demographic and other palliative care data. The Hosmer-Lemshow goodness test for the regression model's fitness was checked and the multi-collinearity test result with a VIF ranged from 1.042 to 1.619.

## Operational definitions

**Palliative care service utilization:** Respondents who scored above median level for 8 outcome measuring variables were considered as adequate utilization, while those who scored equals and or below median level were considered to inadequate utilization [20,21].

**Knowledge about Palliative care.** Based on 13 PaCKS questions; if a respondent answers $\geq$ 50% questions correctly, considered as adequate knowledge and if a respondent answers <50% questions correctly, considered as inadequate knowledge [**18**].

**Attitude to Palliative care.** based on the patients response to the 5 attitude questions; if a respondent answers above median level, considered as good attitude and if a respondent answers equals and or below median level, considered as poor attitude [17].

**Satisfaction.** based on the patient's response to the 8 satisfaction questions; if patients' response above median level were considered to satisfied, and if patients' response equals and or below median level considered to be unsatisfied.

**Severity of pain.** based on WHO pain scale classification [22];

- No Pain–pain Free

- Mild Pain–Nagging, annoying, but doesn't really interfere with daily living activities.

- Moderate Pain–Interferes significantly with daily living activities.

- Severe Pain–Disabling & unable to perform daily living activities.

## Ethical consideration

The proposal was presented to Debre Berhan University Asrat Weldeyes Health Science Campus. A supportive letter and an ethical clearance paper from the Deber-Berehan University Aserat Woldeyese Health Science campus prior to the data gathering period were granted and submitted to the cancer center hospital's administrative offices. Then those institutions gave me permission to collect data. The responders' dignity and rights were respected as well. Each respondent's verbal consent was obtained prior to any data collection, and because medical histories are sensitive personal matters, the importance of maintaining confidentiality was explained.

## Result

### Predisposing factors

**Sociodemographic characteristics.** A total of 404 cancer patients agreed to take part in the survey and completed the questionnaires, with a response rate of 97.56%. 63.4% of the participants in the study were between the ages of 18 and 47. The major portion of participants was married (73.8%), females (56.7%) & had a college or university degree (44.8%), while 12% of them were Unable to read & write. Most of participants 63.4% resided in urban (Table 1).

### Health belief factors (knowledge, attitude)

**Knowledge.** Less than half (48.5%) of participants believed that the goal of palliative care was to address psychological issues caused by an incurable disease. Only 21.3% of patients thought that palliative care was gave exclusively for people who were six months or closer to passing away. While 83.2% thought that the hospital was the only place where palliative care could be accessed (Table 2).

**Attitude.** Participants who have good attitude towards palliative care service utilization were 35.1%. 36.9% were not frightened to consider using palliative care, and 89.4% of them

**Table 1. Socio-demographic characteristics of adult cancer patients receiving palliative care services in Tikur Anbessa Specialized Hospital and Saint Paul's Hospital Millennium Medical College, Addiss Abeba, Ethiopia (n = 404).**

| Characteristics | | Frequency | Percentage |
|---|---|---|---|
| Age | 18–47 | 256 | 63.4 |
| | 48–63 | 103 | 25.5 |
| | ≥64 | 45 | 11.1 |
| Gender | Female | 229 | 56.7 |
| | Male | 175 | 43.3 |
| Residence | Rural | 148 | 36.6 |
| | Urban | 256 | 63.4 |
| Marital status | Single | 60 | 14.8 |
| | Married | 298 | 73.8 |
| | Divorced | 21 | 5.2 |
| | Widowed | 25 | 6.2 |
| Family size | 1–2 | 57 | 14.1 |
| | 3–4 | 201 | 49.8 |
| | ≥ 5 | 146 | 36.1 |
| Education Level | Unable to read & write | 49 | 12.1 |
| | Primary school | 75 | 18.6 |
| | Secondary school | 99 | 24.5 |
| | College/university | 181 | 44.8 |
| Religion | Orthodox | 181 | 44.8 |
| | Protestant | 85 | 21.0 |
| | Muslim | 118 | 29.2 |
| | Other | 20 | 5.0 |

**Table 2. Knowledge status of adult Cancer patients towards palliative care service utilization in Tikur Anbessa Specialized Hospital and Saint Paul's Hospital Millennium Medical College, Addiss Abeba, Ethiopia (n = 404).**

| Characteristics | | Frequency | Percentage |
|---|---|---|---|
| A goal of Palliative Care is to address any psychological issues brought up by serious illness. | Yes | 196 | 48.5 |
| | No | 208 | 51.5 |
| Stress from serious illness can be addressed by palliative care | Yes | 189 | 46.8 |
| | No | 215 | 53.2 |
| Palliative Care can help people manage the side effects of their medical treatment | Yes | 349 | 86.4 |
| | No | 55 | 13.6 |
| When people receive Palliative Care, they must give up their other doctor | Yes | 169 | 41.8 |
| | No | 235 | 58.2 |
| Palliative Care is exclusively for people who are in the last six months of life | Yes | 86 | 21.3 |
| | No | 318 | 78.7 |
| Palliative care is specifically for people with cancer | Yes | 241 | 59.7 |
| | No | 163 | 40.3 |
| People must be in the hospital to receive palliative care | Yes | 336 | 83.2 |
| | No | 68 | 16.8 |
| Palliative care is designed specifically for older adults | Yes | 202 | 50.0 |
| | No | 202 | 50.0 |
| Palliative care is a team-based approach to care | Yes | 316 | 78.2 |
| | No | 88 | 21.8 |
| A goal of Palliative Care is to help people better understand their treatment options | Yes | 394 | 97.5 |
| | No | 10 | 2.5 |
| Palliative care encourages people to stop treatments aimed at curing their illness | Yes | 5 | 1.2 |
| | No | 399 | 98.8 |
| A goal of palliative care is to improve a person's ability to participate in daily activities | Yes | 379 | 93.8 |
| | No | 25 | 6.2 |
| Palliative care helps the whole family cope with a serious illness | Yes | 191 | 47.3 |
| | No | 213 | 52.7 |

**Table 3. Attitude status of adult Cancer patients towards palliative care service utilization in Tikur Anbessa Specialized Hospital and Saint Paul's Hospital Millennium Medical College, Addiss Abeba, Ethiopia (n = 404).**

| Characteristics | Frequency | Percentage |
|---|---|---|
| Cancer patients afraid even to think about Palliative care utilization | | |
| Good | 149 | 36.9 |
| Poor | 255 | 63.1 |
| The thought of Palliative care utilization scares. | | |
| Good | 237 | 58.7 |
| Poor | 167 | 41.3 |
| Cancer patients want to use Palliative care services. | | |
| Good | 361 | 89.4 |
| Poor | 43 | 10.6 |
| Cancer patients have a close relationship with their Palliative care health service providers | | |
| Good | 264 | 65.3 |
| Poor | 140 | 34.7 |
| PC cannot be delivered concurrently with curative cancer treatments | | |
| Good | 212 | 52.5 |
| Poor | 192 | 47.5 |
| Attitude towards Palliative Care Service Utilization | | |
| Good | 142 | 35.1 |
| Poor | 262 | 64.9 |

wish to use palliative care services. From the study participants, 52.5% were not believed to give PC in conjunction with curative cancer treatments (Table 3).

## Health need factors

**Cancer site.** The most common type of cancer among the participants was breast cancer, affecting 30.9% of them (see Fig 3).

**Severity of pain and treatment side effects.** The majority of them reported mild to moderate pain. A 70.8% of cancer patients had side effects from their treatments. Among the treatment side effects, gastro intestinal disturbance (anorexia, nausea, vomiting, and diarrhea) account for 21% (Table 4).

## Enabling factors

**Occupation, monthly income & satisfaction.** The participants' average monthly income was <3001 Ethiopian Birr, and 22.5% of them worked for the government. Most cancer patients (68.6%) have had satisfaction. The majority of them (91.6%) were satisfied with the counseling services, although the majorities were not with the service brochure (97%), reception conditions & recreation room (80.4%), telephone help, and cancer advisory (89.4%). Access to information satisfied 69.6% of participants, but home nursing care and charitable support left 100% respectively. About 97.3% of cancer patients said they were satisfied with their family's support (Tables 5 and 6).

**Health system factors.** According to the responses from the participants, 72.5% of them revealed that hospitals had bureaucratic procedures for accessing palliative care, and just 13.6% said that all medications and procedures were readily available. Ninety-nine percent of patients were required to purchase their prescribed medicines outside of hospitals, and the majority of patients (79.7%) had trouble paying their hospital bills. But 81.2% of the patients received enough time and attention from the service providers (Table 7).

## Prevalence of Palliative care service utilization (PCSU)

The prevalence of palliative care utilization is 35.4% [95% CI: 59.7, 68.6%] (see Fig 4).

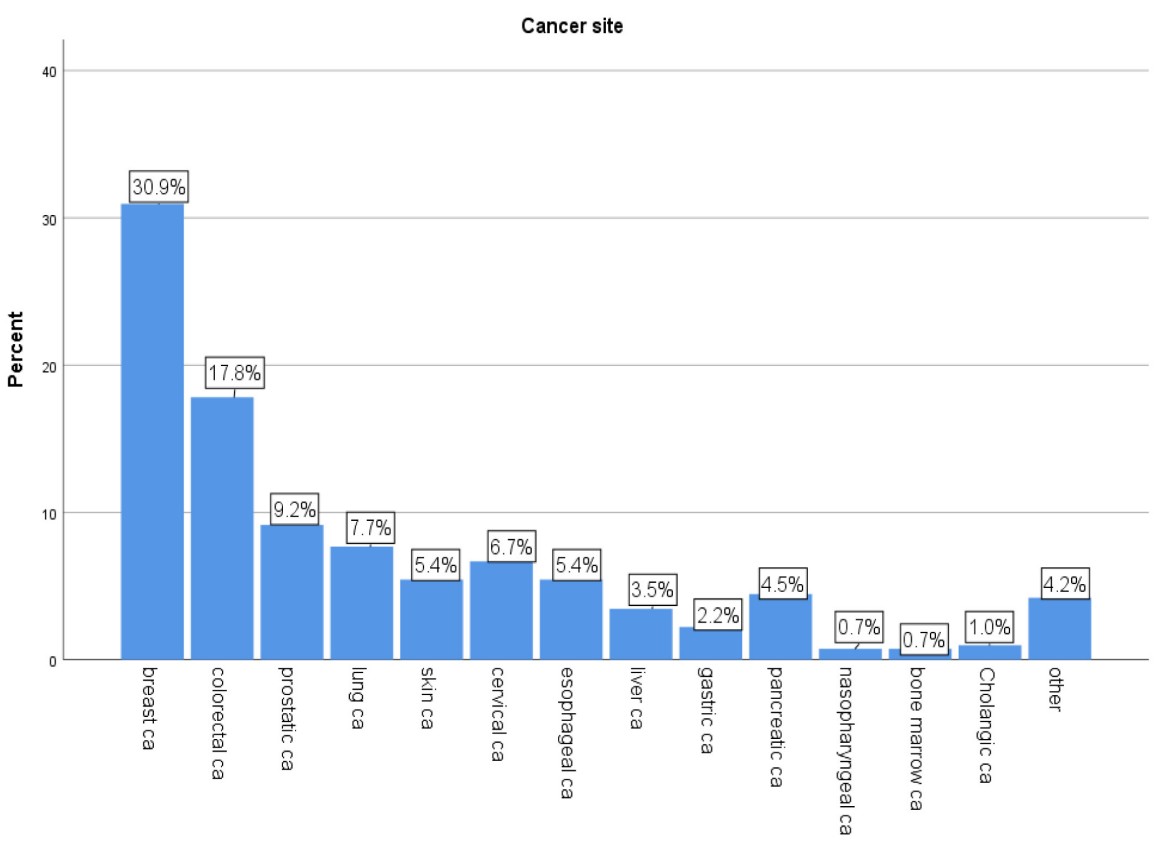

**Fig 3. Cancer site of Patients received palliative care service utilization in Tikure Anbessa specialized hospital and saint paul;s hospital in Addis Abeba, Ethiopia.**

Most cancer patients used chemotherapy (75%) and physical/pain relievers (77.2%). Those participants who used radiotherapy, surgery, psychological support, spiritual support, and financial support were 30%, 20.5%, 17.8%, 17.6%, and 0.7%, respectively. None of them were used for home nursing care (Table 8).

**Table 4. Severity of pain & treatment side effects of adult cancer patients receiving palliative care services in Tikur Anbessa Specialized Hospital and Saint Paul's Hospital Millennium Medical College, Addiss Abeba, Ethiopia (n = 404).**

| Characteristics | | Frequency | Percent |
|---|---|---|---|
| Rate of pain scale | No pain | 60 | 14.9 |
| | Mild pain | 124 | 30.7 |
| | Moderate pain | 146 | 36.1 |
| | Severe pain | 74 | 18.3 |
| Treatment side effects | Gastrointestinal disturbance | 86 | 21.3 |
| | Skin problems | 63 | 15.6 |
| | Central nevus system manifestation | 63 | 15.6 |
| | Loss of hair | 46 | 11.4 |
| | Muscle & joint stiffness | 28 | 6.9 |
| | Total patients treatment side effect experienced | 286 | 70.8 |

GI disturbance = anorexia, nausea, vomiting, and diarrhea, Skin problems = Bruising/ Itching/ Sore mouth, CNS manifestation = Tiredness, Headache.

**Table 5. Jobs held by adult cancer patients and their monthly wages in Tikur Anbessa Specialized Hospital and Saint Paul's Hospital Millennium Medical College, Addiss Abeba, Ethiopia (n = 404).**

| Variables | Frequency | Percentage |
|---|---|---|
| **Occupation** | | |
| Governmental employee | 91 | 22.5 |
| Farmer | 97 | 24.0 |
| Merchant | 63 | 15.6 |
| Other | 153 | 37.9 |
| **Monthly income** | | |
| <3001 | 187 | 46.3 |
| 3001–6000 | 105 | 26.0 |
| >6000 | 112 | 27.7 |

## Factors associated with palliative care service utilization

From eight variables; client education level, occupation, treatment side effects, loss of appetite, distance, availability of all medications & procedures, facing financial shortage and satisfaction found to have p value < 0.25 in the bivariate analysis, only four variables (client education level, treatment side effects, distance, and satisfaction) were found statistically significant (p value < 0.05) via the multivariable logistic regression model and strongly linked to respondents' palliative care service utilization.

Cancer diagnosed patients who have college or university level of education were 2.3 times more likely to use palliative care services than Cancer diagnosed patients with unable to read and write (AOR = 2.3, 95% CI: 1.01, 5.16).

**Table 6. Satisfaction of adult Cancer patient with institutional services and family support in Tikur Anbessa Specialized Hospital and Saint Paul's Hospital Millennium Medical College, Addiss Abeba, Ethiopia (n = 404).**

| Variables | Frequency | Percentage |
|---|---|---|
| Counseling services | | |
| Satisfied | 370 | 91.6 |
| Unsatisfied | 34 | 8.4 |
| Service brochure & benefit | | |
| Satisfied | 12 | 3.0 |
| Unsatisfied | 392 | 97.0 |
| Reception conditions & recreation room | | |
| Satisfied | 79 | 19.6 |
| Unsatisfied | 325 | 80.4 |
| Telephone support and cancer advisory | | |
| Satisfied | 43 | 10.6 |
| Unsatisfied | 361 | 89.4 |
| Access to information | | |
| Satisfied | 281 | 69.6 |
| Unsatisfied | 123 | 30.4 |
| Home nursing service | | |
| Satisfied | 0 | 0 |
| Unsatisfied | 404 | 100.0 |
| Charity support | | |
| Satisfied | 54 | 13.4 |
| Unsatisfied | 350 | 86.6 |
| Family support | | |
| Satisfied | 393 | 97.3 |
| Unsatisfied | 11 | 2.7 |
| Over all patients satisfaction | | |
| High satisfaction | 277 | 68.6 |
| Low satisfaction | 127 | 31.4 |

**Table 7. Responses of adult cancer patients to health system factor questions in Tikur Anbessa Specialized Hospital and Saint Paul's Hospital Millennium Medical College, Addiss Abeba, Ethiopia (n = 404).**

| Variables | N | % |
|---|---|---|
| The hospital has bureaucratic procedures for receiving palliative care | | |
| Yes | 293 | 72.5 |
| No | 111 | 27.5 |
| All medications and procedures available | | |
| Yes | 55 | 13.6 |
| No | 349 | 86.4 |
| Patients are facing financial shortage for hospitalization fee | | |
| Yes | 322 | 79.7 |
| No | 82 | 20.3 |
| Patients are forced to buy prescribed medications outside due to stock-outs | | |
| Yes | 400 | 99.0 |
| No | 4 | 1.0 |
| The service providers give enough time and attention to their patients | | |
| Yes | 328 | 81.2 |
| No | 76 | 18.8 |

Cancer diagnosed patients who did not experienced a side effect from their treatment were 3.5 times more likely to use palliative care services than those who experienced treatment side effects (AOR = 3.5, 95% CI: 2.01, 6.21). Cancer diagnosed patients who traveled less than 23 km (within the area of Addis Ababa) were 1.8 times more likely than those who traveled long distance to use PC services (AOR = 1.8, 95% CI: 1.08, 3.15). Cancer diagnosed patients who satisfied by health care services were 2.1 times more likely to use palliative care services than less-satisfied cancer diagnosed patients (AOR = 2.1, 95% CI: 1.28, 3.59). On the other hand, unavailability of medications and procedures, patients' financial shortage for hospitalization

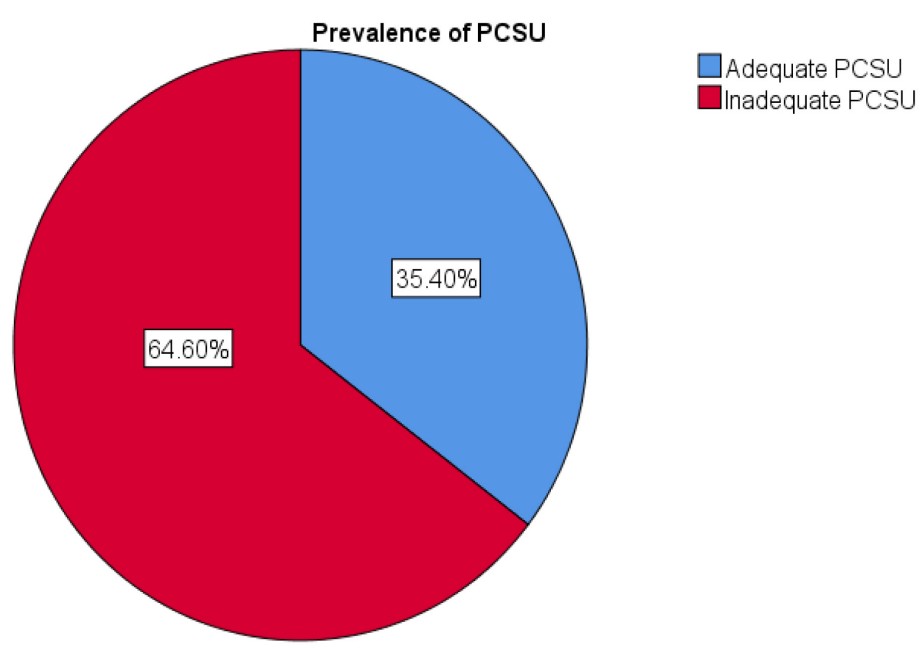

**Fig 4. Palliative care service utilization status among adult cancer patients in Tikur anbessa specialized hospital and Saint Paul's Hospital Millennium Medical College Addiss Abeba, Ethiopia.**

**Table 8. Responses of adult cancer patients to outcome measuring variables of palliative care service utilization in Tikur Anbessa Specialized Hospital and Saint Paul's Hospital Millennium Medical College, Addiss Abeba, Ethiopia (n = 404).**

| Variables | Frequency | Percentage |
|---|---|---|
| Have you received chemotherapy? | | |
| Yes | 303 | 75 |
| No | 101 | 25 |
| Have you received radiotherapy? | | |
| Yes | 121 | 30 |
| No | 283 | 70 |
| Have you received surgical treatment? | | |
| Yes | 83 | 20.5 |
| No | 321 | 79.5 |
| Have you received psychological/emotional support? | | |
| Yes | 72 | 17.8 |
| No | 332 | 82.2 |
| Have you received physical/pain relievers? | | |
| Yes | 312 | 77.2 |
| No | 92 | 22.8 |
| Have you received spiritual support? | | |
| Yes | 71 | 17.6 |
| No | 333 | 82.4 |
| Have you received financial support? | | |
| Yes | 3 | 0.7 |
| No | 401 | 99.3 |

fee, and loss of appetite had an association with PCSU in binary regression but not significantly associated by adjusted odds ratio (Table 9).

## Discussion

The results of this study are in accordance with new and current national, international, and sub-Saharan African (SSA) literature on the variables influencing PC service use among patients diagnosed with cancer. The extent of palliative care service utilization in this study was 35.4% (95% CI: 31.4, 40.3%). This study was a little bit more than previous studies conducted in Asia (35.0%) and less than other studies conducted in the United States (41.9%) and in Ethiopia (TASH) (57.2%) [23–25].

The higher results in the study area might be due to a difference in study population, area, and setup. The study in Asia employed a wide area (i.e., Bangladesh, Philippines, Sri Lanka, and Vietnam) and the rate used was among those with PC awareness (n = 234). The lower results in the study area also might be due to a difference in study population, area, setup, and design. In the United States, the study design was retrospective and National Inpatient Sample data (2005–2014) was used. In the previous TASH, the source of the population was only from one cancer oncology centre, and a smaller sample size was used.

In this study, patients who were college or university education were significantly associated with the presence and use of PC service. Patients diagnosed with cancer who have college or university education were 2.3 times more likely to use palliative care services than those who unable to read and write (AOR = 2.3, 95% CI: 1.01, 5.16). This may be explained by the fact that people with higher level of education easily understand the written and oral instructions given by health care professionals, following instructions like prescriptions or appointment schedules, and understanding the health care system well enough to obtain needed services [26]. Patients with higher educational levels thus frequently receive sufficient health care

**Table 9. Factors associated with Palliative Care Service Utilization at Tikur Anbessa Specialized Hospital and Saint Paul's Hospital Millennium Medical College, Addis Ababa, Ethiopia (n = 404).**

| Variables | Palliative Care Service Utilization | | COR (95% CI) | AOR (95% CI) | P-value |
|---|---|---|---|---|---|
| | Inadequate N (%) | Adequate N (%) | | | |
| Education | | | | | |
| Unable to read & write | 38 (69) | 17(31) | 1 | | |
| Primary school | 49 (65) | 26(35) | 1.2 (0.56, 2.50) | 1.5 (0.69, 3.48) | 0.295 |
| Secondary school | 63 (72) | 25(28) | 0.9 (0.43, 1.85) | 1.8 (0.79, 4.24) | 0.152 |
| Secondary school | 111(60) | 75(40) | 1.5 (0.79, 2.87)** | 2.3 (1.01, 5.16) | 0.046* |
| Occupation | | | | | |
| Other*** | 99(62) | 60(38) | 1 | | |
| Farmer | 63(69) | 28(31) | 0.7 (0.42, 1.27) | 1.9 (0.92, 4.29) | 0.079 |
| Merchant | 47(75) | 16(25) | 0.6 (0.29, 1.08)** | 0.7 (0.34, 1.42) | 0.318 |
| Governmental employee | 52(57) | 39(43) | 1.2 (0.73, 2.09) | 1.3 (0.69, 2.46) | 0.419 |
| Treatment side effect | | | | | |
| Yes | 210(73) | 93(31) | 1 | | |
| No | 51(43) | 67(57) | 3.6 (2.32, 5.68)** | 3.5 (2.01, 6.21) | <0.001* |
| Loss of Appetite | | | | | |
| Yes | 208(69) | 93(31) | 1 | | |
| No | 53(51) | 50(49) | 2.1 (1.34, 3.33)** | 1.2 (0.62, 2.16) | 0.641 |
| Distance | | | | | |
| >23.0 km | 130(73) | 48(27) | 1 | | |
| <23.1 km | 131(58) | 95(42) | 1.9 (1.28, 3.00)** | 1.8 (1.08, 3.15) | 0.024* |
| All medications & procedures available | | | | | |
| Poor(Not available) | 209(69) | 93(31) | 1 | | |
| Good(Available) | 52(51) | 50(49) | 2.2 (1.37, 3.42)** | 1.4 (0.72, 2.60) | 0.334 |
| Patients are facing financial shortage for hospitalization fee | | | | | |
| Poor(Yes) | 188(69) | 83(31) | 1 | | |
| Good(No) | 73(55) | 60(45) | 1.8 (1.21, 2.86)** | 1.1 (0.61, 2.04) | 0.726 |
| Satisfaction | | | | | |
| Low satisfaction | 94(74) | 33(26) | 1 | | |
| High satisfaction | 167(60) | 110(40) | 1.8 (1.18, 2.98)** | 2.1 (1.28, 3.59) | 0.004* |

Other*** = (daily labor, private employee, unable to work because of age and/illness)

** indicates at bi-variables p = < 0.25

*indicates the variables were significant at P<0.05, COR = Crude odds ratio, AOR = Adjusted odds ratio, CI = confidence interval, **1** = reference group (those less to utilize palliative care service were considered as a reference group).

services. Public education was also mentioned in the Korean study as a social need to counter-act misconceptions and cultural attitudes concerning PCSU [27]. Therefore, cancer patients who are college or university education may regularly engage appointments, and follow their prescribed treatment plans, all of which might have an influence to use PC services. This factor was different from earlier findings of TASH in Ethiopia, which showed that formal education was associated with a 49% less likely to utilized palliative care services (AOR = 0.51, 95% CI: 0.23, 0.94) [25]. The reason for this discrepancy may be related to the different sample sizes and numbers of cancer centers included.

The second factor linked to the utilization of PC services was side effects from cancer therapy. Cancer patients who had no side effects from their therapy were 3.5 times more likely to utilize PC services than those who experienced treatment side effects (AOR = 3.5, 95% CI: 2.01, 6.21). No other studies were found on the relationship in this regard. This might be because unfavorable side effects such as bleeding, loss of appetite, diarrhea, exhaustion, hair loss, infections, anemia, sore mouth, and the serious side effect of neutropenia, which may be related to drugs and radiation relatively low in those participants [28,29]. On the other hand, patients from near palliative care center areas with high educational levels are more likely to engage appointments because they understand the side effects that are addressed by PC services. Because of this, those cancer patients did not experiencing side effects from their treatment receive adequate PC services.

Respondents who came from a distance of <23 km were 1.8 times more likely to use palliative care service compared to those who came from a distance of >23.1 km away from PC service centers (AOR = 1.8, 95% CI: 1.08, 3.15). This study was in agreement with the Italian study and Zimbabwe. In Italy, patients were more likely to use palliative care services if they lived less than 20 km from a facility that provided specialist palliative care [30]. Patients in Zimbabwe, who lived in closed distance from radiotherapy facilities had a higher likelihood of receiving palliative radiotherapy [31]. According to a qualitative study conducted at the Parirenyatwa Hospital in Harare, the majority of cancer patients travel great distances to receive care, which results in high transportation costs [32]. So living closest to palliative care centers could be attributed to minimize financial shortage for transportation and hospital service fees, have an access of transportation, and may have good awareness about palliative care services.

This study also showed that respondents who had high satisfaction were 2.1 times more likely to use palliative care services compared to those who had low satisfaction(AOR = 2.1, 95% CI: 1.28, 3.59). Similarly the study in South Carolina shows that patients were highly satisfied with the care they received in the multidisciplinary breast clinic (MDBC) program more likely utilize palliative care service (AOR = 3.77, 95% CI: 3.65, 3.89) [33]. On the other hand; the PC department's appointments and the admission office's waiting time, both of which were considered to be particularly low, were factors in the high satisfaction which increase PCSU [34]. Due to the limits of cross-sectional studies, it may not be possible to define specific qualitative criteria that are associated with PCSU from this finding. This study also did not specify which kind of medication the participants received had any particular side effects. Private cancer palliative care facilities were not also included.

## Conclusion

The extent of palliative care service utilization were lower than previous studies in Ethiopia. Clients' higher educational level, treatment side effects, distance to a medical facility, and high patient satisfaction were all significantly associated with palliative care service utilization. Health care providers working in palliative care centers should improve health education and counseling about PCSU as well as early detection and management of treatment side effects to enhance the patients' quality of life until the end of their life. The Ministry of health should plan for the accessibility of cancer palliative care service centers outside of Addis Ababa, as well as to enhance social support. Researchers should investigate all cancer centers in Ethiopia at a national level through a variety of methods from various parties involved in the delivery of palliative care services in order to identify the factors linked to the use of such services.

### Consent for data collection

During as per international standard or university standard, patient's written consent has been collected and preserved by the data collectors.

## Supporting information

**S1 File. English version questioners.**
(DOCX)

## Acknowledgments

First, we want to express our heartfelt gratitude to Debre Birhan University Asrat Woldeyes Health Science Campus School of Nursing & Midwifery for giving valuable support. Secondly, the deepest gratitude to all Pre-Publication Support Service (PRESS) of PLOS ONE, the staff member of both Saint Paul's Hospital Millennium Medical College and Tikur Anbesa Specialized hospitals, study participants and the data collectors their cooperation in this study.

## Author Contributions

**Conceptualization:** Nigus Afessa, Mitiku Tefera.

**Data curation:** Nigus Afessa, Mitiku Tefera.

**Formal analysis:** Nigus Afessa, Mitiku Tefera.

**Funding acquisition:** Nigus Afessa, Mitiku Tefera.

**Investigation:** Nigus Afessa, Mitiku Tefera.

**Methodology:** Nigus Afessa, Mitiku Tefera.

**Project administration:** Nigus Afessa, Mitiku Tefera.

**Resources:** Nigus Afessa, Mitiku Tefera.

**Software:** Nigus Afessa, Dagmawit Birhanu, Belete Negese, Mitiku Tefera.

**Supervision:** Dagmawit Birhanu, Belete Negese.

**Validation:** Dagmawit Birhanu, Belete Negese.

**Visualization:** Dagmawit Birhanu, Belete Negese.

**Writing – original draft:** Dagmawit Birhanu, Belete Negese.

**Writing – review & editing:** Nigus Afessa, Dagmawit Birhanu, Belete Negese, Mitiku Tefera.

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
