## [Decision Letter · Decision Letter 0]

17 Apr 2023

PONE-D-23-02089

Factors affecting palliative care service utilization among cancer patients at public hospitals oncology unit Addis Ababa, Ethiopia, 2022

PLOS ONE

Dear Dr. Afessa,

Thank you for submitting your manuscript to PLOS ONE. After careful consideration, we feel that it has merit but does not fully meet PLOS ONE’s publication criteria as it currently stands. Therefore, we invite you to submit a revised version of the manuscript that addresses the points raised during the review process.

Generally, the introduction is not aligned to the topic presented. Mainly, the authors are expected to cover the status of palliative care utilization, what should have been done, the gap noted and why the current study has been intended in the specified context. The extended literature on the prevalence of cancer shown in the middle of this section should be limited.The objectives of the study should be integrated in the last section of the introduction.The source and study population is not clearly stated. Please revise. The term ‘cancer patient/s’ is not sounding and appropriate. Please use proper terms, such as ‘patients diagnosed with cancer’.The study appears to assume that every patient diagnosed with cancer should utilize palliative care service. How can this be possible, especially, at the early stages of the disease, or do you have any justification for this?Line 103 ‘’ Those critically ill patients who can’t response during the data collection period were excluded’’. Please check statement for correctness and punctuation. These are patients who actually require PC, but excluded. How can this bias be accounted for?Lines 114 and 116: ‘every 2’  and ‘3rd’ please revise.How was the systematic random sampling methods performed in two different hospitals?Please make tables consistentLine 314 the statement ‘Eight variables were found to have p value < 0.025 in the bivariate analysis. Four variables were found statistically significant via the multivariable logistic regression model’ is not clear or requires revision.Line 319: ‘….were 2.3 times AOR = 2.3, 95% CI (1.01, 5.16)…’ please check and fix rephrase statement appropriately.Lines 321 to 324; ‘’ AOR = 1.8, 95% CI (1.08, 3.15) indicates that respondents who traveled less than 23 km (within the area of Addis Ababa) were 1.8 times more likely than those who traveled long distance to use PC services. High satisfied respondents were 2.1 times (AOR = 2.1, 95% CI (1.28, 3.59)’’ require revision. Please rephrase text of the first statement and appropriate use of the parentheses in the second statement.  Line 417: ‘’heath healthcare’’ please check and fix. Also check uniformity of writing /punctuation under this subsection.Discussion section: flow of ideas seems confusing. The authors started a deductive approach with influencing factors followed by the prevalence, which also has been repeated latter. Please avoid the intermingled narration and follow the logical order. The term ‘prevalence of utilization’ should be replaced by appropriate words/phrases, such as ‘level or extent of utilization’’ as it often connotes negative outcomes.

We look forward to receiving your revised manuscript.

Kind regards,

Tariku Shimels, B. Pharm., B.A, M.Sc.

Academic Editor

PLOS ONE

Journal Requirements:

 “NO”

5. Please upload a copy of Figures 3 and 5 to which you refer in your text on pages 11 and 22. If the figure is no longer to be included as part of the submission please remove all reference to it within the text.

Reviewers' comments:

Reviewer's Responses to Questions

**Comments to the Author**

1. Is the manuscript technically sound, and do the data support the conclusions?

Reviewer #1: Yes

Reviewer #2: No

2. Has the statistical analysis been performed appropriately and rigorously? 

Reviewer #1: Yes

Reviewer #2: Yes

3. Have the authors made all data underlying the findings in their manuscript fully available?

Reviewer #1: Yes

Reviewer #2: No

4. Is the manuscript presented in an intelligible fashion and written in standard English?

Reviewer #1: Yes

Reviewer #2: No

5. Review Comments to the Author

Reviewer #1: Reviewer Comment

Comment 1 (Abstract): Revise and re-write the Background makes it complete paragraph which can introduce the novice and scientific readers about palliative care service utilization.

Comment 2: In general the paper reads better. However, I comment you on the hard work. You have shown the prevalence of the palliative care service utilization as worldwide, at the continent level and the country level. That sounds good, but you didn’t show your research gap well.

Comment 3: The introduction build clearly into the research question. Nevertheless, the research

question is only expressed in the abstract, and should ideally be the last sentence of the

last paragraph.

Comment 4: Some grammatical and language errors exist throughout the article. Please have a

language expert or editor review the article as a whole.

Comment 5: "The checklist was pretested on 5% of cancer patients at Dessie referral hospital for validity and reliability "- please indicate the findings of the pilot mentioned here.

Comment 6 (Data quality management): “The questionnaire was properly designed and written in English first, then translated into Amharic by language experts”- what about other patients who were unable to hear Amharic language?

- The rest of the methodology appears clear

Comment 7 (Ethical Considerations): The study was conducted at TASH and SPHMMC oncology unit, Addis Ababa Ethiopia, but Ethical Review was obtained from Debre-Berhan University! Why? How?

Comment 8 (Results): I am not clear with indicators and parameters the author/s used to measure palliative care service utilization! This the most critical and confusing. It will be difficult to get sound scientific answers for basic research questions in the current forms of result descriptions! What is the importance of identifying factors associated with palliative care service utilization?

Comment 9: The discussion brings the results out and clearly seeks to explain the factors associated with the results found in this study. But a large portion of this discussion is then spent on comparing the results with other studies from Africa and abroad, and commenting that the difference might be due to study population, area, and setup.

Please review this as it makes the discussion elaborate and the reader loses his/her train of thought comparing statistics with all the various countries mentioned.

Conclusion: Well written and concise summary of the article.

Please, would the author/s consider the following questions:

• “What findings from this study are unique?"

• "Why is the clinically relevant?"

• "Would this change practice?"

• "What are next steps?"

Reviewer #2: I suggest authors that this paper could have outcome measurement bias. most interview questions were asked as general questions, as Ethiopian patient do not understand it. This could make the conclusion highly biased as outcome measurement bias.

6. PLOS authors have the option to publish the peer review history of their article (what does this mean?). If published, this will include your full peer review and any attached files.

Reviewer #1: No

Reviewer #2: No

---

## [Author Response · Author response to Decision Letter 0]

7 Jul 2023

We try to respond your comments at all and we ready for your next suggestion.

---

## [Decision Letter · Decision Letter 1]

25 Aug 2023

PONE-D-23-02089R1Factors affecting palliative care service utilization among cancer patients at public hospitals oncology unit Addis Ababa, Ethiopia, 2022PLOS ONE

Dear Dr. Tefera,

Thank you for submitting your manuscript to PLOS ONE. After careful consideration, we feel that it has merit but does not fully meet PLOS ONE’s publication criteria as it currently stands. Therefore, we invite you to submit a revised version of the manuscript that addresses the points raised during the review process.

ACADEMIC EDITOR:Please find the below comments and make sure to address both reviewers' concerns in your revision. Alternatively, attachment of both reviews results has been appended with this decision. Please ensure that your decision is justified on PLOS ONE’s publication criteria and not, for example, on novelty or perceived impact.

We look forward to receiving your revised manuscript.

Kind regards,

Tariku Shimels, B. Pharm., B.A, M.Sc.

Academic Editor

PLOS ONE

Additional Editor Comments:

i) There is mix of using uppercase and lower case words in the abstract and the document. for example, ‘’Occupation, treatment side effects, Loss of Appetite, distance, availability of all medications & procedures, …’ Please revise appropriately.

ii) There is inconsistency in using punctuation for statements presented in the tables (2 and 3). Ideally, you do not need to put a period for such statements.

iii) There is inconsistent use of punctuation, and improper parentheses under the variables subsection. Please revise.

iv) Table 9 needs revision for variables occupation status.

v) The references should be checked against PLOS ONE’s guideline, and styles applied consistently.

Reviewers' comments:

Reviewer's Responses to Questions

**Comments to the Author**

1. If the authors have adequately addressed your comments raised in a previous round of review and you feel that this manuscript is now acceptable for publication, you may indicate that here to bypass the “Comments to the Author” section, enter your conflict of interest statement in the “Confidential to Editor” section, and submit your "Accept" recommendation.

Reviewer #1: All comments have been addressed

2. Is the manuscript technically sound, and do the data support the conclusions?

Reviewer #1: Yes

3. Has the statistical analysis been performed appropriately and rigorously? 

Reviewer #1: Yes

4. Have the authors made all data underlying the findings in their manuscript fully available?

Reviewer #1: Yes

5. Is the manuscript presented in an intelligible fashion and written in standard English?

Reviewer #1: Yes

6. Review Comments to the Author

Reviewer #1: Thank for the rigorous change made to your manuscript. Other comments you need to correct were uploaded.

Reviewer 2 comments

Minor

Iine 159 is not clear. “The Hosmer-Lemshow goodness test for the regression 160 model's fitness was 86.2 %”. What does Hosmer-lemeshow goodness test mean? It is a significance test, but not as mentioned in the text. See it again.Sampling estimation not clear. No citations to former proportion.Why all adults included? Does all cancer patients need palliative care? Why? There will also be easily cured patients…what does PC mean?Why using mean or above for utilization of PC? No any standard in nursing palliative care?Line 274, table 6, why not you present findings in a scale, so that your audience understands where the response has been concentrated.Language editing is big issue??

Major

Line 301, table8, outcome measurement is not satisfactory. It could have been better to ask client speicifically to the services rather than asking for general questions, such as ‘Have you received psychological/emotional support?’, ‘Have you received spiritual support?’, ‘Have you received financial support?’, etc. so, my conclusion is I have no confidence to accept the output with this somewhat general measurements. Because our client doesn’t understand it. Please see it again.**********

7. PLOS authors have the option to publish the peer review history of their article (what does this mean?). If published, this will include your full peer review and any attached files.

Reviewer #1: No

---

## [Author Response · Author response to Decision Letter 1]

29 Aug 2023

Dear editors and reviewers, We have tried to process, edit, and respond to your feedback without wasting any time. Therefore, we are still ready to receive and edit your feedback.

We Thank you very much!

---

## [Editor Report · Decision Letter 2]

1 Sep 2023

PONE-D-23-02089R2Factors affecting palliative care service utilization among cancer patients at public hospitals oncology unit Addis Ababa, Ethiopia, 2022PLOS ONE

Dear Dr. Tefera,

Thank you for submitting your manuscript to PLOS ONE. After careful consideration, we feel that it has merit but does not fully meet PLOS ONE’s publication criteria as it currently stands. Therefore, we invite you to submit a revised version of the manuscript that addresses the points raised during the review process.

ACADEMIC EDITOR:

Still more issues to address: please edit on the following and return.

The first statement of the abstract is complex. Please fragment it into simpler sentences.The statement ‘’….404 of 414’’ in the abstract is confusing. Please state the number only to whom the questionnaires were administered.  Please check some terminologies used, for example what is ‘assistante in mentioning your rank’?Please do not start a paragraph with a number, ex. 78% in the introduction. Also check all text all through the manuscript.There is an unclear punctuation period in citations 11, 13 extra, please check and fix.Last paragraph of the background section, [show fig 1] is not clear. Please either remove ‘show’ or replace by ‘see’.  The same correction needs to be made in other legends (ex. figure 2 or figure 3.In the label ‘schematic presentation….’of the conceptual framework, please add a comma after ‘Addis Ababa’. Make similar corrections in other sections including the methods. Also remove the period at the end of the caption, and add a period after ‘Figure 1’.  Please check PLOS ONE’s criteria of labeling captions.  The phase ‘non cooperative ‘ in the methods should be replaced by ‘non-cooperative’Please check the consistency of in text citation for tables and legends. For example, table 1 and other tables are cited differently (**Table :1** vs. Table 2 or 3). Would you check such mistakes throughout?Please avoid use of a period at the end of all table and figure legends.Table 9 ‘’ Factors associated with Palliative Care Service Utilization shows that Bi-Variable & multivariable logistic regression with Crud & Adjusted Odd Ratio in Tikur Anbessa Specialized Hospital and Saint Paul’s Hospital Millennium Medical College, Ethiopia, 2022 (n = 404).’’There are many problems in this title. I suggest revising this as ‘Factors associated with Palliative Care Service Utilization at Tikur Anbessa Specialized Hospital and Saint Paul’s Hospital Millennium Medical College,  Addis Ababa, Ethiopia, 2022 (n = 404)’’Table 9. There was an earlier comment regarding the variable of ‘occupation status’.  Probably, ‘government employee is in the upper row. Please check and fix.Please include the ‘limitation’ section to the end of the discussion’ preceded by any possible strength.Present the statements under the recommendation section to the end of the conclusion section.

We look forward to receiving your revised manuscript.

Kind regards,

Tariku Shimels, B. Pharm., B.A, M.Sc.

Academic Editor

PLOS ONE

Journal Requirements:

Additional Editor Comments:

Still more issues to address: please edit on the following and return.

i. The first statement of the abstract is complex. Please fragment it into simpler sentences.

ii. The statement ‘’….404 of 414’’ in the abstract is confusing. Please state the number only to whom the questionnaires were administered.

iii. Please check some terminologies used, for example what is ‘assistante in mentioning your rank’?

iv. Please do not start a paragraph with a number, ex. 78% in the introduction. Also check all text all through the manuscript.

v. There is an unclear punctuation period in citations 11, 13 extra, please check and fix.

vi. Last paragraph of the background section, [show fig 1] is not clear. Please either remove ‘show’ or replace by ‘see’. The same correction needs to be made in other legends (ex. figure 2 or figure 3.

vii. In the label ‘schematic presentation….’of the conceptual framework, please add a comma after ‘Addis Ababa’. Make similar corrections in other sections including the methods. Also remove the period at the end of the caption, and add a period after ‘Figure 1’. Please check PLOS ONE’s criteria of labeling captions.

viii. The phase ‘non cooperative ‘ in the methods should be replaced by ‘non-cooperative’

ix. Please check the consistency of in text citation for tables and legends. For example, table 1 and other tables are cited differently (Table :1 vs. Table 2 or 3). Would you check such mistakes throughout?

x. Please avoid use of a period at the end of all table and figure legends.

xi. Table 9 ‘’ Factors associated with Palliative Care Service Utilization shows that Bi-Variable & multivariable logistic regression with Crud & Adjusted Odd Ratio in Tikur Anbessa Specialized Hospital and Saint Paul’s Hospital Millennium Medical College, Ethiopia, 2022 (n = 404).’’There are many problems in this title. I suggest revising this as ‘Factors associated with Palliative Care Service Utilization at Tikur Anbessa Specialized Hospital and Saint Paul’s Hospital Millennium Medical College, Addis Ababa, Ethiopia, 2022 (n = 404)’’

xii. Table 9. There was an earlier comment regarding the variable of ‘occupation status’. Probably, ‘government employee is in the upper row. Please check and fix.

xiii. Please include the ‘limitation’ section to the end of the discussion’ preceded by any possible strength.

xiv. Present the statements under the recommendation section to the end of the conclusion section.

---

## [Author Response · Author response to Decision Letter 2]

2 Sep 2023

Dear Reviewers and Editors, We are very grateful for your additional relevant comments, questions and suggestions and we are suspenseing by your next decision and also comments. 

Thanks to much for your quick response !

---

## [Editor Report · Decision Letter 3]

6 Sep 2023

PONE-D-23-02089R3Factors affecting palliative care service utilization among cancer patients at public hospitals oncology unit Addis Ababa, Ethiopia, 2022PLOS ONE

Dear Dr. Tefera,

Thank you for submitting your manuscript to PLOS ONE. After careful consideration, we feel that it has merit but does not fully meet PLOS ONE’s publication criteria as it currently stands. Therefore, we invite you to submit a revised version of the manuscript that addresses the points raised during the review process.

ACADEMIC EDITOR:Please check comments below.

We look forward to receiving your revised manuscript.

Kind regards,

Tariku Shimels, B. Pharm., B.A, M.Sc.

Academic Editor

PLOS ONE

Journal Requirements:

Additional Editor Comments:

Dear author(s),

Thank you for your revised submission. The manuscript has, now, improved markedly.  However, there are a few issues left unaddressed.

i) The limitation section should appear under the end of the discussion. 

ii) The recommended statements should be linked to the end of the conclusion section. 

More concerns to add:

i) The title has some issues to fix. Suggested title 'Palliative care service utilization and associated factors among cancer patients at oncology units of public hospitals in Addis Ababa, Ethiopia''

ii) Please also check the superscript placement of authors' information on the first page. If you use 1 as superscript in the authors' list, you should follow the same style in the information provided below the list.

---

## [Author Response · Author response to Decision Letter 3]

6 Sep 2023

We are happy by your comments and suggestion so we thanks too.

---

## [Editor Report · Decision Letter 4]

11 Sep 2023

PONE-D-23-02089R4Factors affecting palliative care service utilization among cancer patients at public hospitals oncology unit Addis Ababa, Ethiopia, 2022PLOS ONE

Dear Dr. Tefera,

Thank you for submitting your manuscript to PLOS ONE. After careful consideration, we feel that it has merit but does not fully meet PLOS ONE’s publication criteria as it currently stands. Therefore, we invite you to submit a revised version of the manuscript that addresses the points raised during the review process.

ACADEMIC EDITOR: Please check a few more comments provided below once again. I would suggest that you go through a published PLOS paper and make all necessary changes before returning this revision. 

We look forward to receiving your revised manuscript.

Kind regards,

Tariku Shimels, B. Pharm., B.A, M.Sc.

Academic Editor

PLOS ONE

Journal Requirements:

Additional Editor Comments:

Dear Author(s),

Thank you for submitting your revision. Unfortunately, the manuscript still suffers from a lot of unaddressed and additional issues.

a)Title: please avoid putting a period at the end of the title. Good also if you avoid the date (2022) in the objective.

b) authors' information: please put as for example this: '' 1Department of Nursing, school of nursing and midwifery, Asrat woldeyes health science campus, Debre Berhan University, Ethiopia'' if all authors are from the same department. Please note that 1 is in superscript form with no dot after it, and that there is comma after 'midwifery'. In addition, please put 1 for all authors if they still belong to the same department. If not, assign them affiliation. Next, remove 2 and add the following instead:

*correspondence: put your email here

c) The discussion section requires revision considering the way the 95% C.Is were stated. Please follow a uniform and meaningful style of presenting it. For example the statement

i) ''Patients diagnosed with cancer who have college or university education were 2.3 times AOR = 2.3, 95% CI (1.01, 5.16) more likely to use palliative care services than those who unable to read and write.'' in the discussion OR

ii) ''Cancer diagnosed patients who traveled less than 23 km (within the area of Addis Ababa) were 1.8 times AOR = 1.8, 95% CI (1.08, 3.15) more likely than those who traveled long distance to use PC services.'' in the results are either meaningless or unclear.

Please revise the results and discussion for consistency and clarity. You could rewrite it, for example, as ''Patients diagnosed with cancer and who have college or university education were 2.3 times more likely to use palliative care services as compared with those who were unable to read and write (AOR = 2.3, 95% CI:1.01, 5.16) for i above. It should be uniform for all cases, however.

iii) The use of the 95% CI is inconsistent. In all sections, authors used square brackets mixed with open brackets. Please use uniform formatting, preferably the one that does not confuse with citations i.e ().

c) It would be advisable to remove the abbreviations section, but make sure to define full term, in brackets at first use.

d)The references are not adequately revised. Some lacked either of a journal name, a publisher, a doi, or a page number when appropriate. The URL name/site and access date should also be mentioned when full journal profile may not be available, for example for repository manuscripts not published. There is also a duplicate in year of publication. Please revise all thoroughly.

---

## [Author Response · Author response to Decision Letter 4]

16 Sep 2023

We are very grateful for your relevant comments, questions, and suggestions, and we are anticipating your next decision and also your comments.

We thanks too!

---

## [Editor Report · Decision Letter 5]

30 Oct 2023

Palliative care service utilization and associated factors among cancer patients at oncology units of public hospitals in Addis Ababa, Ethiopia

PONE-D-23-02089R5

Dear Dr. Tefera,

We’re pleased to inform you that your manuscript has been judged scientifically suitable for publication and will be formally accepted for publication once it meets all outstanding technical requirements.

Kind regards,

Tariku Shimels, B. Pharm., B.A, M.Sc.

Academic Editor

PLOS ONE

Additional Editor Comments (optional):

Please check and fix these issues:

a) Remove the dot (.) from the end of the title 

b) Refer a Plos paper and fix the title page accordingly. For example, no address has been provided for superscript 2. Also put the numbers in the address details as were in the author lists (i.e. as superscript) 
---

## [Editor Report · Acceptance letter]

17 Nov 2023

PONE-D-23-02089R5 

Palliative care service utilization and associated factors among cancer patients at oncology units of public hospitals in Addis Ababa, Ethiopia. 

Dear Dr. Tefera:

I'm pleased to inform you that your manuscript has been deemed suitable for publication in PLOS ONE. Congratulations! Your manuscript is now with our production department. 

Kind regards, 

on behalf of

Mr. Tariku Shimels 

Academic Editor

PLOS ONE